NIPS4Bplus: a richly annotated birdsong audio dataset

Morfi Veronica g.v.morfi@qmul.ac.uk 1
Bas Yves 2 3
Pamuła Hanna 4
Glotin Hervé 5
Stowell Dan 1
1 Machine Listening Lab, Centre for Digital Music (C4DM), Department of Electronic Engineering and Computer Science, Queen Mary University of London , London , United Kingdom
2 Centre d’Ecologie et des Sciences de la Conservation (CESCO), Muséum National d’Histoire Naturelle, CNRS, Sorbonne Université , Paris , France
3 Centre d’Ecologie Fonctionnelle et Evolutive (CEFE), CNRS, Université de Montpellier, Université Paul-Valéry Montpellier , Montpellier , France
4 Department of Mechanics and Vibroacoustics, AGH University of Science and Technology , Kraków , Poland
5 CNRS, LIS, DYNI team, SABIOD, Université de Toulon (UTLN), Aix Marseille Université (AMU) , Marseille , France
Gomez Shawn
Electronic publication date: 2019 Oct 7
Publication date: 2019
Volume: 5
Electronic Location ID: e223
Received 2019 May 1; Accepted 2019 Aug 31
Copyright: ©2019 Morfi et al.
Copyright year: 2019
Copyright holder: Morfi et al.
License: This is an open access article distributed under the terms of the Creative Commons Attribution License, which permits unrestricted use, distribution, reproduction and adaptation in any medium and for any purpose provided that it is properly attributed. For attribution, the original author(s), title, publication source (PeerJ Computer Science) and either DOI or URL of the article must be cited.
License URL: https://creativecommons.org/licenses/by/4.0/

Keywords: Audio dataset, Bird vocalisations, Ecosystems, Ecoacoustics, Rich annotations, Bioinformatics, Audio signal processing, Bioacoustics

Funding: EPSRC fellowship EP/L020505/1 AGH-UST Dean’s 16.16.130.942 NIPS4B challenge EADM MaDICS CNRS ANR-18-CE40-0014 Dan Stowell is supported by EPSRC fellowship EP/L020505/1. Hanna Pamuła is supported by AGH-UST Dean’s Grant number 16.16.130.942. SABIOD MI CNRS provided financial support for the NIPS4B challenge, and EADM MaDICS CNRS provided ANR-18-CE40-0014 SMILES supporting this research.

==============================
Recent advances in birdsong detection and classification have approached a limit due to the lack of fully annotated recordings. In this paper, we present NIPS4Bplus, the first richly annotated birdsong audio dataset, that is comprised of recordings containing bird vocalisations along with their active species tags plus the temporal annotations acquired for them. Statistical information about the recordings, their species specific tags and their temporal annotations are presented along with example uses. NIPS4Bplus could be used in various ecoacoustic tasks, such as training models for bird population monitoring, species classification, birdsong vocalisation detection and classification.

Introduction

The potential applications of automatic species detection and classification of birds from their sounds are many (e.g., ecological research, biodiversity monitoring, archival) (Dawson & Efford, 2009; Lambert & McDonald, 2014; Drake et al., 2016; Sovern et al., 2014; Marques et al., 2012). In recent decades, there has been an increasing amount of ecological audio datasets that have tags assigned to them to indicate the presence or not of a specific bird species. Utilising these datasets and the provided tags, many authors have proposed methods for bird audio detection (Adavanne et al., 2017; Pellegrini, 2017) and bird species classification, e.g., in the context of LifeCLEF classification challenges (Goëau et al., 2016; Goëau et al., 2017) and more (Salamon & Bello, 2017; Knight et al., 2017). However, these methods do not predict any information about the temporal location of each event or the number of its occurrences in a recording.

Some research has been made into using audio tags in order to predict temporal annotations, labels that contain temporal information about the audio events. This is usually done in a multi-instance learning (MIL) or weakly supervised learning setting. In (Briggs et al., 2014; Ruiz-Muñoz, Orozco-Alzate & Castellanos-Dominguez, 2015), the authors try to exploit audio tags in birdsong detection and bird species classification, in (Fanioudakis & Potamitis, 2017), the authors use deep networks to tag the temporal location of active bird vocalisations, while in (Roger et al., 2018), the authors propose a bioacoustic segmentation based on the Hierarchical Dirichlet Process (HDP-HMM) to infer song units in birdsong recordings. Furthermore, some methods for temporal predictions by using tags have been proposed for other types of general audio (Schlüter, 2016; Adavanne & Virtanen, 2017; Kumar & Raj, 2016). However, in all the above cases some kind of temporal annotations were used in order to evaluate the performance of the methods. Hence, acquiring temporal annotations is vital even for methods that are in a weakly supervised learning setting.

In the field of automatic birdsong monitoring, advances in birdsong detection and classification have approached a limit due to the lack of fully annotated datasets. Annotating ecological data with temporal annotations to train sound event detectors and classifiers is a time consuming task involving a lot of manual labour and expert annotators. There is a high diversity of animal vocalisations, both in the types of the basic syllables and in the way they are combined (S. Brandes, 2008; Kroodsma, 2005). Also, there is noise present in most habitats, and many bird communities contain multiple bird species that can potentially have overlapping vocalizations (Luther, 2008; Luther & Wiley, 2009; Pacifici, Simons & Pollock, 2008). These factors make detailed annotations laborious to gather, while on the other hand acquiring audio tags takes much less time and effort, since the annotator has to only mark the active sound event classes in a recording and not their exact boundaries. This means that many ecological datasets lack temporal annotations of bird vocalisations even though they are vital to the training of automated methods that predict the temporal annotations which could potentially solve the issue of needing a human annotator.

Recently, BirdVox-full-night (Lostanlen et al., 2018), a dataset containing some temporal and frequency information about flight calls of nocturnally migrating birds, was released. However, BirdVox-full-night only focuses on avian flight calls, a specific type of bird calls, that usually have a very short duration in time. The temporal annotations provided for them do not include any onset, offset or information about the duration of the calls, they simply contain a single time marker at which the flight call is active. Additionally, there is no distinction between the different bird species, hence no specific species annotations are provided, but only the presence of flight calls through the duration of a recording is denoted. Hence, the dataset can provide data to train models for flight call detection but is not efficient for models performing both event detection and classification for a variety of bird vocalisations.

In this paper, we introduce NIPS4Bplus, the first ecological audio dataset that contains bird species tags and temporal annotations. NIPS4Bplus contains temporal annotations for the recordings that comprised the training set of the 2013 Neural Information Processing Scaled for Bioacoustics (NIPS4B) challenge for bird song classification (http://sabiod.univ-tln.fr/nips4b/challenge1.html) that are accessible at Figshare (https://doi.org/10.6084/m9.figshare.6798548) (Morfi, Stowell & Pamuła, 2018). NIPS4Bplus can be used for training supervised automated methods that perform bird vocalisation detection and classification and can also be used for evaluating methods that use only audio tags or no annotations for training.

Table 1 presents an overview comparison between NIPS4plus and the most recent and frequently used datasets in tasks related to bird vocalisation classification and detection. During the 2018 Detection and Classifiaction of Acoustic Scenes and Events, the (DCASE) challenge the freefield1010, warblrb10k, BirdVox-DCASE-20k (deriving from BirdVox-full-night (https://wp.nyu.edu/birdvox/birdvox-full-night/)), Chernobyl and PolandNFC datasets were used in task 3 for bird audio detection, namely detecting the presence of any bird in a recording and assigning a file format with the presence of any bird in a recording and assigning a binary label (1:bird, 0:no-bird) to it (http://dcase.community/challenge2018/task-bird-audio-detection). Another very widely known challenge that addresses the task of active bird species identification in a recording is BirdClef, which has been part of the LifeClef challenge since 2014 (https://www.imageclef.org/lifeclef2019). Finally, BirdVox-full-night presented in (Lostanlen et al., 2018), is a dataset of ten hours of night calls annotated as single points in time instead of continuous events, due to the short duration of night calls in the dataset. The datasets used in BirdClef derive from xeno-canto, the largest publicly available bird sound database, that contains over 450,000 recordings of more than 10,000 different bird species (https://www.xeno-canto.org/). Another bird sound database presented in (Arriaga et al., 2015), that has been open to the public since 2015, is Bird-DB (http://taylor0.biology.ucla.edu/birdDBQuery/). Bird-DB consists of more than 800 recordings from over 30 different bird species. In contrast to xeno-canto that only provides tags of the recordings with the bird species present in it, the recordings in Bird-DB include temporal annotations identifying the bird species and also classifying the vocalisation. Even though Bird-DB provides temporal annotation, it is meant to be used as a database and is not very convenient as a dataset. This is mainly due to the fact that any user can upload recordings and their annotations, additionally, each recording and annotation pair needs to be downloaded separately.

Table 1 Some of the latest and most frequently used datasets in tasks related to bird song classification and detection.

#recs denotes the number of recordings in the dataset; #classes denotes the numbers of classes in each dataset; species tags indicates if there are species specific labels in the recordings stating the presence of specific species in them; annotations denotes the presence of temporal annotations in recordings; duration denotes the approximate duration of each dataset in hours; and other info provides additional information about the characteristics of the dataset.

Dataset Name	#recs	#classes	species tags	annotations	duration	other info	
NIPS4Bplus	687	87	Yes	Yes	1 h		
freefield1010	7,690	N/A	No	No	21 h	bird/no-bird tags	
warblrb10k	10,000	N/A	No	No	28 h	bird/no-bird tags	
BirdVox-DCASE-20k	20,000	N/A	No	No	55 h	bird/no-bird tags	
Chernobyl	6,620	N/A	No	No	18 h	bird/no-bird tags	
PolandNFC	4,000	N/A	No	No	11 h	bird/no-bird tags	
LifeClef(BirdClef) 2019	50,000	659	Yes	No	350 h	from xeno-canto	
LifeClef(BirdClef) 2018	48,843	1500	Yes	No	68 h	from xeno-canto	
BirdVox-full-night	6	25	No	Yes	60 h	points in time	

The rest of the paper is structured as follows: Audio Data Collection describes the process of collecting and selecting the recordings comprising the dataset, Annotations presents our approach of acquiring the tags and temporal annotations and provides statistical information about the labels and recordings comprising the dataset followed by Example Uses of NIPS4Bplus and Conclusion.

Audio Data Collection

The recordings that comprise the NIPS4B 2013 training and testing dataset were collected by recorders placed in 39 different locations, which can be summarised by seven regions in France and Spain. Twenty percent of the recordings were collected from the Haute-Loire region in Central France, 65% of them were collected from the Pyrénées-Orientales, Aude and Hérault regions in south-central France along the Mediterranean cost and the remaining 15% of the recordings originated from the Granada, Jaén and Almeria regions in eastern Andalusia, Spain. The Haute-Loire area is a more hilly and cold region, while the rest of the regions are mostly along the Mediterranean coast and have a more Mediterranean climate.

The recorders used were the SM2BAT (https://bit.ly/2RBf1cd) using SMX-US microphones (https://www.wildlifeacoustics.com/images/pdfs/UltrasonicMicrophones.pdf), both produced by Wildlife Acoustics (https://www.wildlifeacoustics.com/). They were originally put in the field for bat echolocation call sampling, but they were also set to record for 3 h single channel at 44.1 kHz sampling rate starting 30 min after sunrise, right after bat sampling. The recorders were set to a 6 dB Signal-to-Noise Ratio (SNR) trigger with a window of 2 s, and acquired recordings only when the trigger was activated.

Approximately 30 h of field recordings were collected. Any recording longer than 5 s was split into multiple 5 s files. SonoChiro, a chirp detection tool used for bat vocalisation detection, was used on each file to identify recordings with bird vocalisations (http://www.leclub-biotope.com/fr/72-sonochiro). A stratified random sampling was then applied to all acquired recordings, based on locations and clustering of features, to maximise the diversity in the labelled dataset, resulting in nearly 5,000 files being chosen. Following the first stage of selection, manual annotations were produced for the classes active in these 5,000 files and any recordings that contained unidentified species’ vocalisations were discarded. Furthermore, the training set and testing set recordings were allocated so that the same species were active in both. Finally, for training purposes, only species that could be covered by at least seven recordings in the training set were included in the final dataset, the rest were considered rare species’ occurrences that would make it hard to train any classifier; hence, they were discarded. The final training and testing set consist of 687 files of total duration of less than an hour, and 1,000 files of total duration of nearly two hours, respectively.

Annotations

Tags

The labels for the species active in each recording of the training set were initially created for the NIPS4B 2013 bird song classification challenge (Glotin et al., 2013). There is a total of 51 different bird species active in the dataset. For some species we discriminate the song from the call and from the drum. We also include some species living with these birds: nine insects and an amphibian. This tagging process resulted in 87 classes. A detailed list of the class names and their corresponding species English and scientific names can be found in (Morfi, Stowell & Pamuła, 2018). These tags only provide information about the species active in a recording and do not include any temporal information. In addition to the recordings containing bird vocalisations, some training files only contain background noise acquired from the same regions and have no bird song in them, these files can be used to tune a model during training. Figure 1 depicts the number of occurrences per class for recordings collected in each of the three different general regions of Spain, South France and Central France. Each tag is represented by at least seven up to a maximum of 20 recordings.

Figure 1 Label occurrences on different regions.

Number of occurrences of each sound type in recordings collected from Spain, Southern France and Central France.

Each recording that contains bird vocalisations includes one to six individual labels. These files may contain different vocalisations from the same species and also may contain a variety of other species that vocalise along with this species. Figure 2 depicts the distribution of the number of active classes in the dataset.

Figure 2 Number of active classes throughout the dataset.

Distribution of number of active classes in dataset recordings.

Figure 3 depicts the number of co-occurrences between pairs of labels. We can notice that there are no notable patterns to the ways species vocalisations co-occur. One interesting thing one can notice while studying the co-occurrence heat map is that there is no strong correlation between calls and songs from the same species, this is due to the different functions between calls and songs produced. As calls may be related to self-maintenance activities such as species identification or holding the flock together, while songs are mostly used for attracting a mate, establishing territories, intimidating enemies and learning through imitations and practising.

Figure 3 Label co-occurrence heat map.

Distribution of number of active classes in dataset recordings.

Temporal annotations

Temporal annotations for each recording in the training set of the NIPS4B dataset were produced manually using Sonic Visualiser (https://www.sonicvisualiser.org/). The temporal annotations were made by a single annotator, Hanna Pamuła, and can be found in (Morfi, Stowell & Pamuła, 2018). Table 2 presents the temporal annotation format as is provided in NIPS4Bplus.

In Fig. 4 we present the mean duration for every class activation in all the recordings. Most classes have a brief duration of less than 0.5 s, with most of the insect classes (marked with red bars) having a longer duration. Finally, in Fig. 5 we report the total number of activations for each class in the dataset, with the minimum being 1 and the maximum being 282.

Table 2 An example of NIPS4Bplus temporal annotations.

Starting Time (sec)	Duration (sec)	Tag	
0.00	0.37	Serser_call	
0.00	2.62	Ptehey_song	
1.77	0.06	Carcar_call	
1.86	0.07	Carcar_call	
2.02	0.41	Serser_call	
3.87	1.09	Ptehey_song	

In concern to the temporal annotations for the dataset, we should mention the following:

• The original tags were used for guidance; however, some files were judged to have a different set of species than the ones given in the original metadata. Similarly, in a few rare occurrences, despite the tags suggesting a bird species active in a recording, the annotator was not able to detect any bird vocalisation.

• An extra ‘Unknown’ tag was added to the dataset for vocalisations that could not be classified to a class.

• An extra ‘Human’ tag was added to a few recordings that have very obvious human sounds, such as speech, present in them.

• Out of the 687 recordings of the training set 100 recordings contain only background noise, hence no temporal annotations were needed for them.

• Of the remaining 587 recordings that contain vocalisations, six could not be unambiguously labelled due to hard to identify vocalisations, thus no temporal annotation files were produced for them.

• An annotation file for any recording containing multiple insects does not differentiate between the insect species and the ‘Unknown’ label was given to all insect species present.

• In the rare case where no birds were active along with the insects no annotation file was provided. Hence, seven recordings containing only insects were left unlabelled.

• In total, 13 recordings have no temporal annotation files. These can be used when training a model that does not use temporal annotations.

• On some occasions, the different syllables of a song were separated in time into different events while in other occasions they were summarised into a larger event, according to the judgement of the expert annotator. This variety could help train an unbiased model regarding separating events or grouping them together as one continuous time event.

As mentioned above, each recording may contain multiple species vocalising at the same time. This can often occur in wildlife recordings and is important to be taken into account when training a model. Fig. 6 presents the fraction of the total duration containing overlapping vocalisations as well as the number of simultaneously occurring classes.

Figure 4 Mean value and standard deviation of the duration of each class in NIPS4Bplus in seconds.

Blue bars indicate bird label, red bars indicate insect label and green indicate amphibian.

Figure 5 Total number of each class activations in NIPS4Bplus.

Blue bars indicate bird label, red bars indicate insect label and green indicate amphibian.

Figure 6 Number of simultaneous active classes over the total duration of the data.

Distribution of simultaneous number of active classes on the total duration of the recordings.

Example Uses of NIPS4Bplus

A few examples of the NIPS4Bplus dataset and temporal annotations being used can be found in (Morfi & Stowell, 2018a) and (Morfi & Stowell, 2018b). First, in (Morfi & Stowell, 2018a), we use NIPS4Bplus to carry out the training and evaluation of a newly proposed multi-instance learning (MIL) loss function for audio event detection. And in (Morfi & Stowell, 2018b), we combine the proposed method of (Morfi & Stowell, 2018a) and a network trained on the NIPS4Bplus tags that performs audio tagging in a multi-task learning (MTL) setting.

For both experiments, we split the NIPS4B training dataset into a training set and a testing set. For our training set, the first 499 recordings of the NIPS4B training dataset are used, while the rest are included in our testing set, excluding 14 recordings for which confident strong annotations could not be attained. Those 14 recordings are added to our training set resulting to a grand total of 513 training recordings and 174 testing recordings. Out of the 513 training recordings a small subset of them are used during training for validation purposes only. More specifically, the validation set consists of 63 recordings (55 containing at least one bird vocalisation, 8 without any vocalisation), with the rest 450 recordings (385 containing at least one bird vocalisation, 65 without any vocalisation) used only for training the model. Detailed results can be found in Morfi & Stowell (2018a) and Morfi & Stowell (2018b).

Additional applications using NIPS4Bplus could include training models for bird species audio event detection and classification, evaluating how generalisable of method trained on a different set of data is, and many more. More specifically, the dataset and the temporal annotations can be used for evaluating methods that have been trained without temporally annotated data. In general, this kind of data, that lack temporal annotation, can be easily acquired in a large scale which is suitable for training deep learning approaches. However, temporally annotated data are needed to properly evaluate the performance of models that perform their prediction, hence another way of using NIPS4Bplus along with other datasets is as an evaluation set.

Conclusion

In this paper, we present NIPS4Bplus, the first richly annotated birdsong audio dataset. NIPS4Bplus is comprised of the NIPS4B dataset and tags used for the 2013 bird song classification challenge plus the newly acquired temporal annotations. We provide statistical information about the recordings, their species specific tags and their temporal annotations.

We thank Sylvain Vigant for providing recordings from Central France, and BIOTOPE for making the data public for the NIPS4B 2013 bird classification challenge.

Additional Information and Declarations

Competing Interests

Author Contributions

Data Availability

Dan Stowell is an Academic Editor for PeerJ.

Veronica Morfi conceived and designed the experiments, performed the experiments, analyzed the data, contributed reagents/materials/analysis tools, prepared figures and/or tables, performed the computation work, authored or reviewed drafts of the paper, approved the final draft.

Yves Bas, Hanna Pamuła and Hervé Glotin analyzed the data, contributed reagents/materials/analysis tools, authored or reviewed drafts of the paper, approved the final draft.

Dan Stowell conceived and designed the experiments, analyzed the data, contributed reagents/materials/analysis tools, prepared figures and/or tables, authored or reviewed drafts of the paper, approved the final draft.

The following information was supplied regarding data availability:

The data is available at Figshare: Morfi, Veronica; Stowell, Dan; Pamula, Hanna (2018): NIPS4Bplus: Transcriptions of NIPS4B 2013 Bird Challenge Training Dataset. figshare. Dataset. https://doi.org/10.6084/m9.figshare.6798548.

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
