# Peer review of "NIPS4Bplus: a richly annotated birdsong audio dataset"

_PeerJ Computer Science, doi:10.7717/peerj-cs.223_

## Round 0.1 · original submission · Major Revisions

· Academic Editor

Major Revisions

Please address the reviewers concerns. In particular, greater detail on the properties of the data as well as its accessibility are particularly important.

Reviewer 1 ·

Basic reporting

Important issues:

- In the abstract, this dataset is qualified as "the first richly annotated birdsong audio data set." This is a very strong affirmation that is not clearly supported in the manuscript. I suggest adding a section or a table, comparing this dataset with datasets used in previous studies in bioacoustics.

- In lines 53-55, it is said that "many ecological datasets lack temporal annotations of bird vocalisations even though they are vital to the training of automated methods that predict the temporal annotations [...]" This is not always true since some recognition techniques are designed to handle data with imprecise annotations (e.g., multi-instance learning, or weakly supervised learning; as an example, see https://arxiv.org/pdf/1606.03664.pdf). I suggest to the authors mentioning this type of algorithms and how the dataset can help them (they might work better having some data precisely annotated, also, it might help for performance evaluation).

Less important points:

- The origin of the name "NIPS4B" could be explained in the introduction.

Experimental design

- Locations, annotation procedure, and settings are well described.

Validity of the findings

No comments

Additional comments

- In this paper, a fully annotated dataset of birdsong recording is introduced.

- Since the dataset was used in 2013, it would be important to enhance the details that were recently added.

·

Basic reporting

The paper describes a database for bird vocalizations detection and species recognition in audio recordings.
The suggested annotated dataset is useful because the task of localizing vocalizations in the spectrogram for several species is important but there aren’t many available databases around. The paper is well written though it would benefit from a revision.
It is a small database of few hundred of recordings and it will not suffice to train generic bird-recognizers but it is a good start and we hope that it would be enlarged in the future. This paper stands a merit but I suggest the following changes:
1. SM2BAT,SMX-US give details of these products i.e. company, country city.
2. A paper must be self-sufficient. The reader should not have to refer to other papers that have part of the info that this paper needs. Having said that, I suggest you include in the supplement a) the full list of bird species b) the annotation file of the database
3. I cannot read the x-axis legend of Figure 2. The legend of regions at top right falls on the measurements. One way is to rotate by 90o end enlarge. Another way is to include a better quality figure in the supplementary material
4. Provide a link of where the database can be found (most important suggestion). This paper is not of any use to the reader if it is not accompanied by a link pointing to the database and annotation files. I made a quick check and I believe it is available but you need to make clear to the readers of this journal where it is located.

Experimental design

The suggested annotated dataset is useful because the task of localizing vocalizations in the spectrogram for several species is important but there aren’t many available databases around.

Validity of the findings

Please, provide a link of where the database can be found (most important suggestion). This paper is not of any use to the reader if it is not accompanied by a link pointing to the database and annotation files. I made a quick check and I believe it is available but you need to make clear to the readers of this journal where it is located.

Reviewer 3 ·

Basic reporting

This paper reports on a new richly annotated birdsong audio dataset, called NIPS4BPLUS, which has both label and temporal annotation information for each 5-sec short audio recording. The author showed the procedure of data collection and annotation in detail and showed example use of the data set for machine learning which was reported in another paper.

Experimental design

The procedure of data creation is well described. However, I think that the more information about the temporal annotation is beneficial for readers to understand the characteristics of the data set more clearly.
For example, the total song (or call) duration and song count for each species over all recordings can provide basic information of the temporal annotation.

Validity of the findings

As an example use section shows the NIPS4BPLUS can provide a good dataset for machine learning tasks. I think that it would be better to elaborate more on whether and how the newly added temporal annotation data can facilitate learning performance or learning procedures.

In addition, the authors emphasized that this is the first richly annotated birdsong audio dataset. However, the Bird-DB also appears to be a rich data set in that it provides exact temporal annotation information of various species in recordings.

Arriaga JG, Cody ML, Vallejo EE, Taylor CE. 2015. Bird-DB: a database for annotated bird song sequences. Ecol Inform. 27:21–25.

Additional comments

I believe that the paper and the proposed dataset can significantly contribute to birdsong classification studies if some more information is provided as described above.

---

## Round 0.2 · Minor Revisions

· Academic Editor

Minor Revisions

Thank you for addressing the reviewer questions/concerns. There is just one issue that is tangentially brought up by Reviewer 1 that does need clarification. Specifically, it is not clearly obvious where the data is located and it is very easy to not realize that the files and annotations are publicly accessible through a figshare link in the Morfi et al., 2018 reference. Please incorporate language that makes this more clear to the reader (perhaps in the paragraph beginning on line 68 or the very end of the introduction before Audio Data Collection or some other prominent location). For example, maybe something like "The complete set of audio files and associated annotations are accessible at https://doi.org/10.6084/m9.figshare.6798548 (Morfi et al., 2018)" would be much more clear and would certainly help to reduce any confusion regarding access to the data and related files.

It appears that related questions regarding datafile structure are largely addressed at this site, so I do not think this has to be addressed here (though you are welcome to provide additional details if you feel it would be useful). Also, it is up to you whether you wish to include the data set brought up by Reviewer 1 - it is not required.

Reviewer 1 ·

Basic reporting

The authors clearly describe the advantages of their fully annotated dataset of bird vocalizations. They provide references to other bioacoustic datasets and mention the processing and recognition techniques that have been applied to them. I suggest considering, as an additional dataset, this one: https://www.kaggle.com/c/mlsp-2013-birds

It has been used in several publications, like the ones below:

https://www.sciencedirect.com/science/article/pii/S1574954115000102
https://link.springer.com/article/10.1007/s11265-016-1155-0
https://academic.oup.com/biolinnean/article/115/3/731/2440478

Experimental design

In the manuscript, the authors provide some statistics of the dataset. I suggest including also the structure of the folders and the format of the files.

Validity of the findings

My main concern is that the dataset is not available. I suggest to share the dataset and the source code to generate the statistics provided in the manuscript, and some demos to process the recordings.

On line 121, it is said that only species that covered at least 7 recordings were included. I guess that that number wouldn't be enough to train a recognition model.

Additional comments

Since the purpose of this manuscript is to describe a dataset, it should be publically available. I would suggest accepting the manuscript after taking a look at the dataset.

Below are some minor suggestions:

- Line 100: The meaning of the abbreviation should be introduced the first time that is mentioned
- Line 108: Try to avoid the redundancy on the phrase "The recorders used to acquire the recordings"

·

Basic reporting

The authors have answered/corrected all points raised by my review.
As far as it concerns me, the manuscript should be accepted for publication

Experimental design

The authors have answered/corrected all points raised by my review.
As far as it concerns me, the manuscript should be accepted for publication

Validity of the findings

The authors have answered/corrected all points raised by my review.
As far as it concerns me, the manuscript should be accepted for publication

Additional comments

The authors have answered/corrected all points raised by my review.
As far as it concerns me, the manuscript should be accepted for publication

Reviewer 3 ·

Basic reporting

The manuscript was significantly improved according to the reviewers' comments.

Experimental design

The manuscript provided sufficient information about the proposed dataset.

Validity of the findings

The more detailed description of the benefit of the use of the proposed data set was provided.

Additional comments

The manuscript was significantly improved according to the reviewers' comments.
I really appreciate detailed replies to my comments.

---

## Round 0.3 · accepted · Accept

· Academic Editor

Accept

Thank you for the clarifying edits and congratulations again.